# A tightly-bonded and flexible mesoporous zeolite-cotton hybrid hemostat

Lisha Yu[1], Xiaoqiang Shang[1], Hao Chen[1], Liping Xiao[1], Yihan Zhu[2] & Jie Fan [1]

Achieving rapid definitive hemostasis is essential to ensure survival of patients with massive bleeding in pre-hospital care. It is however challenging to develop hemostatic agents or dressings that simultaneously deliver a fast, long-lasting and safe treatment of hemorrhage. Here, we integrate meso-/micro-porosity, blood coagulation and stability into a flexible zeolite-cotton hybrid hemostat. We employ an on-site template-free growth route that tightly binds mesoporous single-crystal chabazite zeolite onto the surface of cotton fibers. This hemostatic material maintains high procoagulant activity after water flow treatment. Chabazite particles are firmly anchored onto the cotton surface with < 1% leaching after 10 min of sonication. The as-synthesized hemostatic device has superior hemostatic performance over most other clay or zeolite-based inorganic hemostats, in terms of higher procoagulant activity, minimized loss of active components and better scalability for practical applications (a hemostatic T-shirt is hereby demonstrated as an example).

[1] Key Lab of Applied Chemistry of Zhejiang Province, Department of Chemistry, Zhejiang University, Hangzhou 310027, China. [2] Department of Chemical Engineering and State Key Laboratory Breeding Base of Green Chemistry Synthesis Technology, Zhejiang University of Technology, Hangzhou 310014, China. Correspondence and requests for materials should be addressed to Y.Z. (email: yihanzhu@zjut.edu.cn) or to J.F. (email: jfan@zju.edu.cn)

D eath from hemorrhage represents a substantial global problem, which results in nearly half of the potential life lost caused by cancer[1]. In daily life, first-aid knowledge suggests to apply direct pressure on the wound or cut with a clean cloth, tissue, or piece of gauze, until bleeding stops. However, these materials cannot arrest hemorrhage threat due to lack of hemostatic functions. Achieving rapid definitive hemostasis for pre-hospital care is essential to ensure survival of patients with massive bleeding. The safety, efficacy, manipulability, cost-effectiveness, manufacturability, and biocompatibility are essential parameters for an ideal hemostat[2,3], which however cannot be achieved at the same time for most hemostatic agents or dressings.

Notably, a few types of inorganic materials, including zeolites and clays, have been developed to accelerate the coagulation of the blood without introducing any biohazardous effects[2-4]. These inorganic hemostats usually function through three major mechanisms: (i) absorbing water from the blood and concentrating the blood components at the hemorrhagic site; (ii) activating the blood coagulation cascade; (iii) providing a physical barrier to blood flow[2-4]. Clays usually possess a layered crystalline aluminosilicate structure, which however unavoidably experience severe shrinking and swelling upon water desorption and absorption. This effect degrades the mechanical strength and largely compromises the stability of clay-based hemostats against the loss of active components. The Combat Gauze (Z-Medica), a clay (kaolin) impregnated gauze, works as a popular hemostat in the US military[3,4]. It however fails to achieve rapid definitive hemostasis due to the high loss of active components and also suffers from potential risks of distal thrombosis arising from wound contamination by the detached clay powder[5]. On the other side, zeolites belong to another family of crystalline aluminosilicates with a typical pore size of 0.4–1.2 nm[6-9], which possess cage-like cavities that can accommodate both water molecules and positively charged ions such as $Ca^{2+}$ and $Na^+$ that further help to entrap more water molecules through electrostatic interactions. These interconnected cavities constitute a rigid and three-dimensional porous network that facilitates the fast diffusion of water molecules without introducing considerable structural deformation. Thus, zeolites offer long-term physical and chemical stability while absorbing large amounts of water molecules and exchanging ions with the surrounding media. In addition, they are cheap and biocompatible without any considerable toxicity. A number of studies showed that some natural zeolites have highly appreciated properties and can be implemented in textile for medical purpose[10-14]. As a granular form of zeolite A, QuikClot (Z-Medica) was used as an effective hemostat, of which the operation procedures are however complicated. These zeolite grains have to be used together with binders that not only dilute zeolite as the active component but also block their pores[15]. After the application of QuikClot, adequate debridement like saline flush is required before the closure of wound[16]. Moreover, this type of zeolitic hemostat also has severe side effects, such as thermal injures. Till now, despite the remarkable advantages of zeolites, the chemical synthesis of zeolite-based hemostatic materials that approaching the ideal hemostat remains challenging.

A successful design of zeolitic hemostat should deliver a fast, long-lasting, and safe treatment of hemorrhage. The design requires: (i) a proper selection of zeolite with a porous network enabling fast diffusion and adsorption of adsorbate molecules; (ii) the zeolite immobilization that allows a long-lasting application and prevents the leaching problem; (iii) a nontoxic and low-cost hemostatic system that is easy to operate. Chabazite-type (CHA) zeolites usually exhibit excellent dehydration activity that is an important property for blood concentration and hemostasis, which is comparable with many other zeolites, such as zeolite A[17,18]. It is well accepted that the outstanding hemostatic performance of microporous zeolites can be attributed to their high microporosity and the large surface area. The introduction of mesoporosity into the zeolites further increases the pore sizes and specific surface areas, interrupts the microporous framework of zeolites and results in shorter diffusion length, all of which together contribute the faster diffusion and absorption of water molecules as well as the more effective concentration of blood components at the hemorrhagic site[7,8,19,20]. Cotton is used as a substrate for the direct growth of mesoporous zeolites, due to its low cost and easy manufacturability by weaving and knotting. The direct on-site growth of zeolite on the cellulose allows the strong adhesion between them through chemical bonding and homogeneous zeolite coatings[21-25]. In contrast to those reported zeolitic composites that only involves weak interactions between zeolites and substrates via, for example, impregnation or physical mixing[26-29], the on-site growth route used here ensures the strong stability, high loading, and minimized leaching of the hybrid hemostat without introducing organic templates and binders.

Herein, we develop an on-site template-free growth route of mesoporous zeolite CHA (mCHA) onto the surface of cotton fibers, which forms a tightly bonded mesoporous zeolite CHA-cotton (mCHA-C) hybrid hemostat. The as-synthesized mCHA-C hybrid material exhibits superior hemostatic activity and outperforms the conventionally used kaolin clay impregnated gauze (Combat Gauze, CG) in terms of rapid definitive hemostasis, easy operation, and minimized side effects. The hybrid hemostat is also flexible and scalable for the facile fabrications of diverse textile-based wearable system for early hemorrhage control in our daily life.

## Results

**Template-free zeolite growth on cotton.** The mCHA-C composite was prepared via an on-site template-free growth, in which cotton was used as a structural scaffold. To induce CHA nucleation onto the cotton substrate, cotton and gel precursor (9 $Na_2O$:0.7 $Al_2O_3$:10 $SiO_2$:331 $H_2O$) derived from mixing colloidal silica and aluminate were constrained to implement direct hydrothermal treatment. As can be seen from powder X-ray diffraction (XRD) patterns (Supplementary Fig. 1a), the reflections can be indexed by chabazite (JCPDS No. 34-0137) and cellulose. The thermogravimetry analysis (TGA) revealed that the average loading content of mCHA on cotton from three different parts was $23.3 \pm 0.8$ wt% (mean ± standard deviation (SD)), indicating a good homogeneity of CHA zeolite distribution (Supplementary Table 1). Nitrogen sorption isotherm of mCHA-C showed a significant increase of the adsorption branch between 0.5 and 0.9 $P/P_0$ and a pronounced hysteresis. This type-IV isotherm was typical for mesoporous materials (Supplementary Fig. 1b). The BET surface area and pore volume of mCHA zeolite were estimated to be 489 $m^2 \cdot g^{-1}$ and 0.08 $cm^3 \cdot g^{-1}$, respectively (the contribution from the cotton support was excluded, Supplementary Table 2).

The size, shape, morphology, and texture of the as-synthesized mCHA-C hybrid hemostat are perfectly inherited from the pristine cotton substrate with high flexibility and manufacturability (Fig. 1a). Field-emission scanning electron microscope (FE-SEM) images clearly show a high loading of well-separated mesoporous CHA zeolites anchored on the cotton fiber, with an average size of 5 μm (Fig. 1b). Homogeneously dispersed and well-isolated CHA zeolites on the surface of cotton are advantageous to inherit the structural flexibility from the cotton fibers, in contrast to the vegetable fibers coated with a conformal surface lining of zeolite[21-25], which would easily break and fall off from the fibers upon bending or twisting, and thus restrict its hemostatic applications. Close inspection of individual mCHA

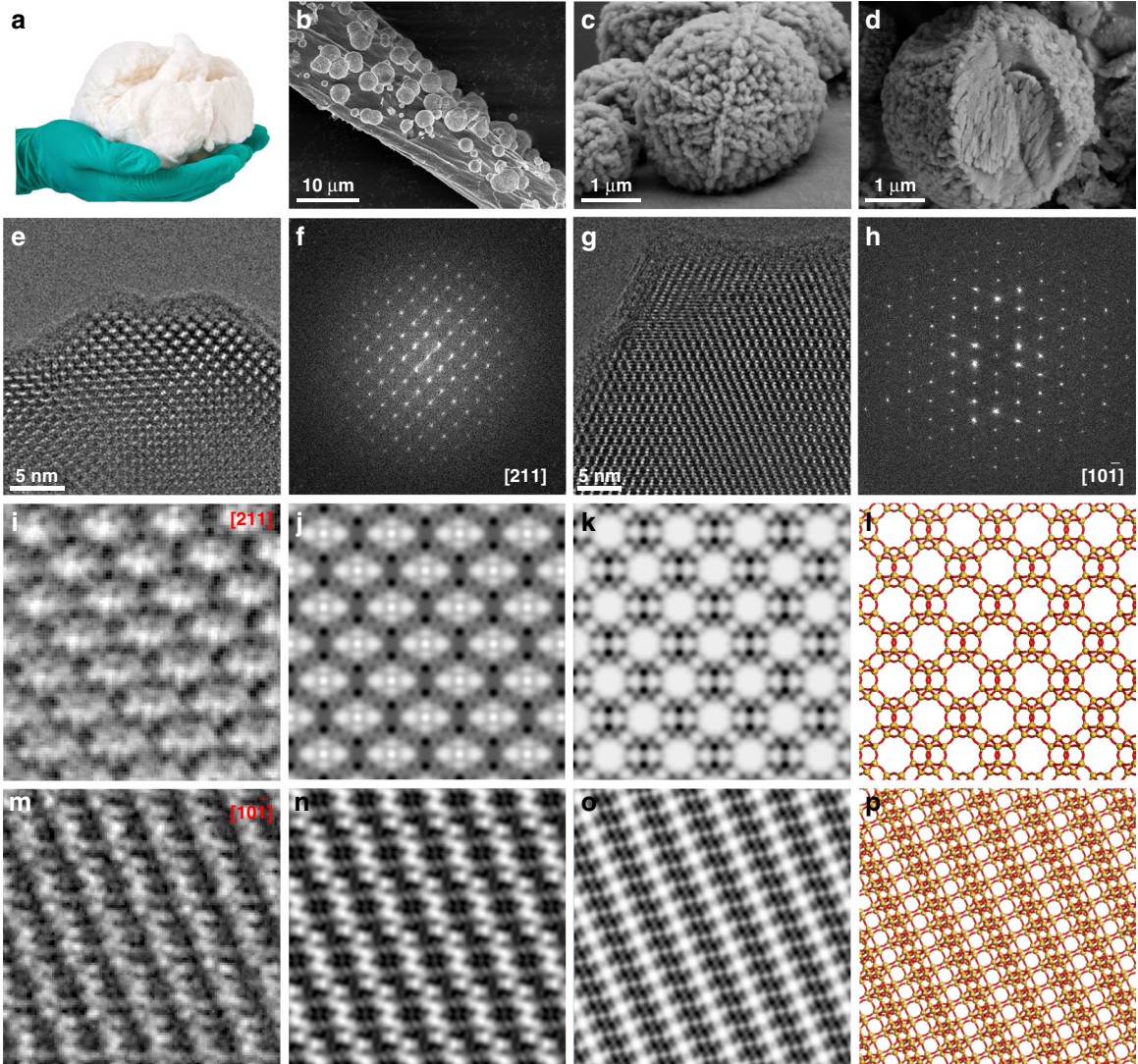

**Fig. 1** Characterization of mCHA-C. **a** Photograph of mCHA-C. FE-SEM images of (**b**) mCHA-C, (**c**) mCHA zeolite on cotton, and (**d**) mCHA zeolite after removing cotton fiber by calcination. HRTEM images and corresponding FFT of individual CHA crystals taken along the (**e**, **f**) [211] and (**g**, **h**) [10$\bar{1}$] zone axes, respectively. **i**, **m** CTF-corrected and denoised images using a Wiener filter at a defocus value of −630 nm and −22 nm, respectively, (**j**, **n**) Cmm and P2 projection symmetry-imposed and lattice-averaged images, where the averaged in-channel contrast in (**j**) might originate from either adsorbate species or CTF effects (**k**, **o**) simulated projected potential maps with a point spread function width of 1.4 Å, (**l**, **p**) projected structural models of the CHA along [211] and [10$\bar{1}$] zone axes, respectively

particles confirms that each particle consists of assembled CHA nanocrystals (148 ± 37 nm; data values corresponded to mean ± SD, $n = 108$; Figure 1c; Supplementary Fig. 2). Notably, the observed interfaces between mCHA particles and cotton fiber are flat and sharp instead of being a small kissing point for most zeolitic hemostats that involve only weak zeolite–substrate interactions[27]. This point is further verified by directly visualizing the zeolite–substrate interface after burning out the cotton substrate (Fig. 1d), which clearly shows a fragmentary spherical shape of mCHA particles with a missing wedge as the interface that binds tightly to the cotton fibers through chemical bonding. The interaction between mCHA zeolite particles and cotton fiber is so strong that free-standing as-grown mCHA zeolite particles (i.e., without heat treatment) can only be collected by using an ultrasonic cell crusher under a high power, for a more detailed microstructural inspection. By using a low-dose transmission electron microscopy (TEM) scheme[30,31], we were able to image the mCHA zeolite without any considerable electron beam

damage. From a low-magnification TEM image, it is clear that the mCHA zeolite particles have surface textures arising from the presence of mesoporosity, as observed by SEM (Supplementary Fig. 3). Electron diffraction patterns collected along two zone axes [211] and [10$\bar{1}$] over a whole mCHA particle confirm the single-crystal nature of the mCHA particles as the oriented-assembly of CHA nanocrystals and can be indexed by the CHA zeolitic structure (Supplementary Fig. 4). The appearance of several reflections along the [10$\bar{1}$] zone axis that should extinct for a perfect CHA structure may originate from the dynamical effect. High-resolution (HR) TEM images taken at a thin protruding edge region of a mCHA particle surface along both [211] and [10$\bar{1}$] directions are shown in Fig. 1e and g. These HRTEM images retain the structural integrity of the mCHA zeolites and have the structural information transfer up to 1.4 Å according to the FFT patterns (Fig. 1f, h). By correcting the defocus effect caused by the contrast-transfer-function (CTF), it is possible to generate chemically interpretable images that match well with the

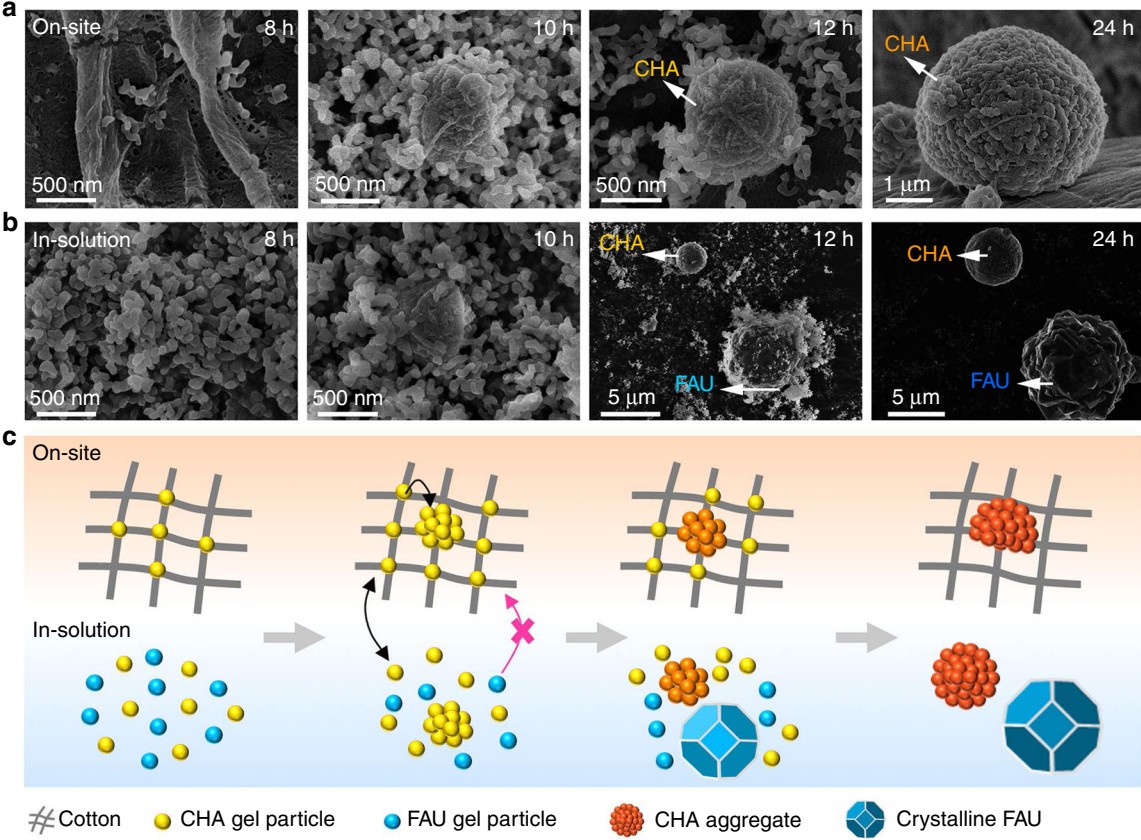

**Fig. 2** Tentative mechanism of mCHA-C formation. FE-SEM images of the products (**a**) on site and (**b**) in solution at different time intervals, (**c**) schematic representation of formation process of on-site and in-solution products

calculated projected potential map and structural projections of the CHA zeolite structure along these two directions (Fig. 1i–p). To the best of our knowledge, this is the first template-free synthesis of mesoporous single-crystal CHA zeolite on cotton, which perfectly retains the physicochemical properties and stability of microporous zeolitic framework while creates mesoporosity by interrupting the framework for greatly enhanced adsorption and diffusion of diverse adsorbates.

**Mechanism of zeolite growth**. To provide mechanistic insights into the growth of mCHA-C, we tracked the entire process of crystal growth, including gel formation, nucleation, and crystallization on the surface of cotton (Fig. 2a; Supplementary Figs. 5, 6). At 8 h, worm-like gel particles with a size of 50–100 nm were observed on the cotton surface, and their amorphous nature was revealed by XRD analysis (Supplementary Fig. 6). Afterwards, the content of these gel particles on the cotton surface (Supplementary Table 3) significantly increased from 5.4 wt% (8 h) to 10.2 wt% (10 h), and started to assemble on the cotton surface. Prolonged hydrothermal treatment up to 12 h led to the formation of partially crystallized spherical particles, which exhibited characteristic (101) and (401) reflections for CHA zeolite in the XRD patterns (Supplementary Fig. 6). The mean coherent length of the crystallites was calculated to be ~7.8 nm using the (401) reflection, which was much smaller than the apparent size of gel particles (50–100 nm, Fig. 2a; Supplementary Fig. 5). It suggests a complex agglomerate with small CHA crystallites embedded into the amorphous matrix, which served as the seeds for further crystallization as described in previous reports[32,33]. The amorphous components were consumed in next steps and fully

crystallized particles of CHA zeolite were obtained between 12 h and 24 h. FE-SEM analysis demonstrated that all individual gel particles disappeared and only spherical aggregates are formed on the surface of cotton after 24 h (Fig. 2a; Supplementary Fig. 5).

In our experiment, the cotton scaffold is critical for the nucleation and growth of CHA zeolite. We found that the on-site grown products were single-phase CHA zeolitic crystals (Fig. 2a; Supplementary Fig. 1a), while the in-solution grown products were a mixture of spherical CHA and bulk faujasite (FAU) zeolitic crystals (Fig. 2b; Supplementary Figs. 7, 8). The FAU zeolites with a lower density (13.3 T atoms per nm³) tend to nucleate and crystallize from those low-density amorphous gel particles in solution. In contrast, the CHA zeolites with a higher density (15.1 T atoms per nm³) tend to nucleate and crystallize from those gel particles directly grown on the cotton surface. These fibers were made of dense-phase silicates, where the silicate lattice puts on a strong constraint and direct the nucleation and crystallization of CHA phase. Besides, the control experiment confirmed that the absence of cotton substrate resulted in a significant loss of CHA structure in the in-solution grown zeolitic products (Supplementary Fig. 9).

To further confirm the key role of the gel particles formed on cotton surface, we added cotton after the formation of gel particles. Dynamic light scattering (DLS) experiment suggested that the gel particles with size of 50–100 nm formed in solution at room temperature for 24 h (Supplementary Fig. 10). To this solution, the cotton was added and then transferred to hydrothermal process. The rest steps were completely identical to the synthesis of mCHA-C. In the end, TGA measurement revealed that there was no zeolite on cotton surface, which was significantly different with mCHA-C. Also, the in-solution

product contained only FAU zeolite crystals (Supplementary Fig. 11), indicating that FAU gel particles seem to be totally incapable of attaching to cotton surface in our experiment (Fig. 2c). The results proved that only the gel particles induced on cotton surface could be further developed to final fiber-bound CHA zeolite. Moreover, the gel particles formed on the cotton surface at the early-stage could further self-assemble and crystalize into mesoporous CHA zeolitic crystals, probably via surface migration (Fig. 2c). In addition, due to the much higher framework density of CHA zeolites compared with the gel particles bonded with cotton fibers, the nucleation and crystallization of CHA particles from these gel particles are accompanied with the concurrent formation of mesopores. Thus, these observations imply that the introduction of cotton surface with highly reactive hydroxyl groups is a key factor for chemical bonding and immobilization of the active mCHA zeolites.

**Stability of composite material.** Fabric-based hemostatic dressing is clinically used hemostatic device with irreplaceable advantages like easy application and excellent shape adaptivity[34]. Among the numerous products, kaolin impregnated CG (Z-Medica) has been proved to be effective in various situations; however, a potential risk arises from the leakage of kaolin powder into the wound[3–5]. In addition, we prepared another zeolite-impregnated sample in which the zeolite suspension was simply dropped onto cotton and then dried (denoted as Im-zeolite-C). Considerable loss of kaolin or zeolite particles was observed after soaking CG and Im-zeolite-C into water (Fig. 3a). Moreover, the stability of hemostat against the water flow is also an important issue, because in practice massive

blood will flow out from the vessel in a traumatic injury. To mimic this condition, we washed the hemostats with deionized water for three times to evaluate the functional benefit of mCHA-C over the other two hemostats. The addition of mCHA-C remarkably accelerated clotting time to $133.0 \pm 2.6$ s (mean $\pm$ SD, $n = 3$) in the plasma clotting assay, compared with the cotton ($353.0 \pm 16.5$ s). The mCHA-C maintained similar procoagulant activity after washing. In contrast, CG or Im-zeolite-C displayed significantly prolonged clotting time after losing significant amount of active hemostatic components (Fig. 3b; Supplementary Fig. 12). We subsequently monitored the mice skin wound after the treatment of Cy5-labeled mCHA-C or CG, using in vivo fluorescence imaging (Fig. 3c). About 36.5% kaolin species were observed to be stripped off from CG substrate and severely contaminate the wound site (calculated by the total fluorescent radiant efficiency, Supplementary Figs. 13–15), whereas no fluorescent signal at the wound site was observed in the mCHA-C group. Note that it is extremely challenging for naked eye to identify and eliminate the kaolin or zeolite nano/microparticular contaminations at the wound site.

To this end, we believe that the binding strength between hemostatic component and matrix should be evaluated in a harsh condition to meet the diverse complicated hemorrhage conditions, during transportation, storage, and application. Sonication works as an effective tool to measure the binding strength as well as the overall mechanical stability of the hybrid hemostat under harsh conditions[15]. As for CG or Im-zeolite-C, the significant leaching was observed right after 1-min sonication (> 80% kaolin or zeolite). Conversely, it was clearly shown that the mCHA particles firmly anchored onto the cotton surface by

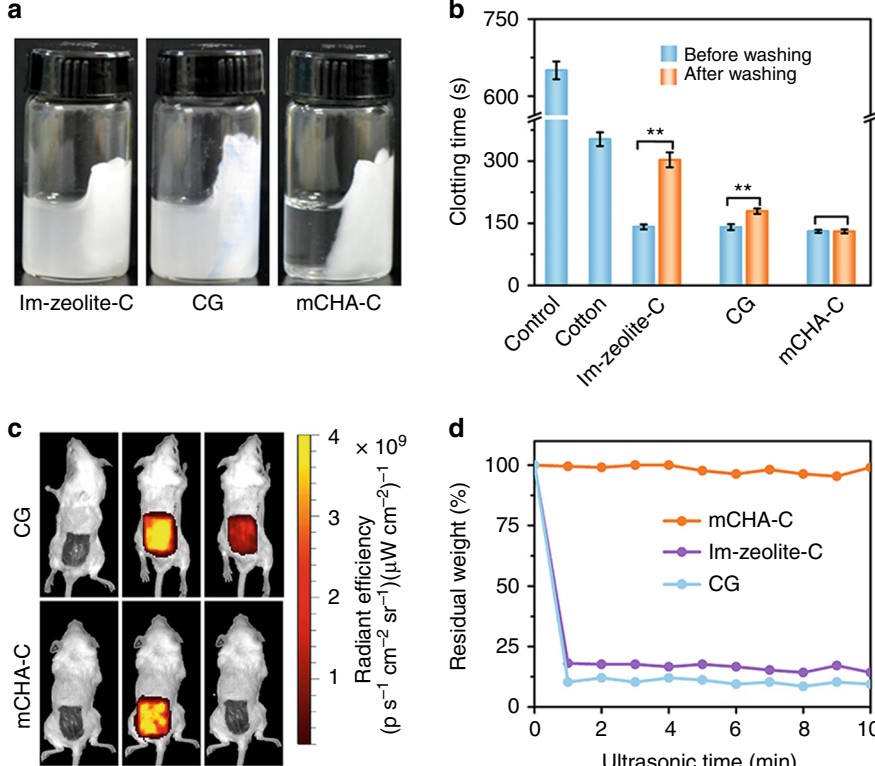

**Fig. 3** Binding strength of hemostatic component and matrix. **a** Photograph of Im-zeolite-C, CG, and mCHA-C after soaking into water. **b** Plasma clotting time of hemostats before and after washing with deionized water. Source data are provided as a Source Data file. Data values corresponded to mean ± standard deviation (SD), $n = 3$. Error bars represent SD. **P < 0.01, one-way analysis of variance (ANOVA). **c** Fluorescence imaging of mice skin wound before (left), during (mid), and after (right) being treated with Cy5-labeled mCHA-C and CG, respectively. **d** The relative residual weight of hemostatic component on mCHA-C (orange), Im-zeolite-C (violet), CG (turquoise) after different ultrasonic time. Source data are provided as a Source Data file

exhibiting < 1% leaching after 10-min sonication (Fig. 3d). This significant difference of binding strength was further supported by evaluating the in vitro procoagulant activity (Supplementary Fig. 16). Consequently, the strong affinity between mCHA and cotton in the mCHA-C hybrid hemostat guarantees uncompromised safety and high procoagulant activity during various practical hemostatic applications.

**Hemostatic evaluation.** The hemostatic performance of mCHA-C was further evaluated in a rabbit lethal femoral artery injury model. First of all, femoral artery was exposed and transected, and the terrible wound gushed with blood (Supplementary Fig. 17). Direct manual pressure was applied on the cut with hemostatic dressing, until bleeding stopped. The addition of mCHA-C accelerated clotting time to $159 \pm 12$ s (mean $\pm$ SD, $n = 8$), compared with the cotton ($383 \pm 56$ s). Moreover, due to a strong affinity between mCHA zeolite and cotton, mCHA-C exhibited superior procoagulant activity (with a much shorter clotting time) over CG ($392 \pm 56$ s) on gushing blood (Fig. 4a). The blood loss in mCHA-C group ($6.1 \pm 1.4$ g) was about 40% less than that in CG group ($10.3 \pm 3.7$ g, Fig. 4b). After hemostasis was achieved by mCHA-C, an open and flat wound surface was presented (Fig. 4c), which gave a better visualization of the surgical area. On the contrary, the animals treated with CG underwent high active component leakage and suffered more blood loss during prolonged bleeding time, which resulted in an extended blood clot that covered all over the injure site (Fig. 4c). During practical hemostatic applications, the massive bleeding would make the loosely attached kaolin leave from near

the injured site, which leads to a low procoagulant performance of hemostatic dressing near the injured site. Besides, there are great chances that loosely attached active component would stick to the tissue around the wound, resulting in indispensable and scrupulous debridement. Instead, the removal of mCHA-C was an easy procedure, and meticulous debridement was dispensable. In addition, mCHA-C overcame the exothermic reaction during the application of zeolite-based granular hemostats[16,35,36] (Supplementary Fig. 18). Moreover, the mCHA-C offered a statistically significant advantage by decreasing the mortality rate to 0% with effective hemostasis and less blood loss (Supplementary Fig. 19).

As a proof-of-concept hemostatic device, we simply scaled up the current synthesis and successfully fabricated a mCHA-based hemostatic T-shirt (denoted as mCHA/T-shirt) with a loading of $21.7 \pm 0.5$ wt% (mean $\pm$ SD, $n = 3$; Fig. 4d; Supplementary Table 4). The good homogeneity of the zeolite immobilization was also demonstrated by in vitro procoagulant activity in five different parts of mCHA/T-shirt (ca. 145 s, Supplementary Fig. 20). Moreover, the mCHA/T-shirt showed an extremely high stability and durability against hand and machine washing or even sonication without losing any noticeable procoagulant activity (Supplementary Fig. 21). Its effectiveness in hemorrhage control, and wearable feature (hygroscopic, soft, breathable, and comfortable to human skin) could provide all-time, instant and affordable protection from accident trauma caused hemorrhage.

In summary, a mesoporous CHA zeolite/cotton hybrid was synthesized in the absence of organic structural directing agents. The CHA gel particles preferably formed on the surface of cotton

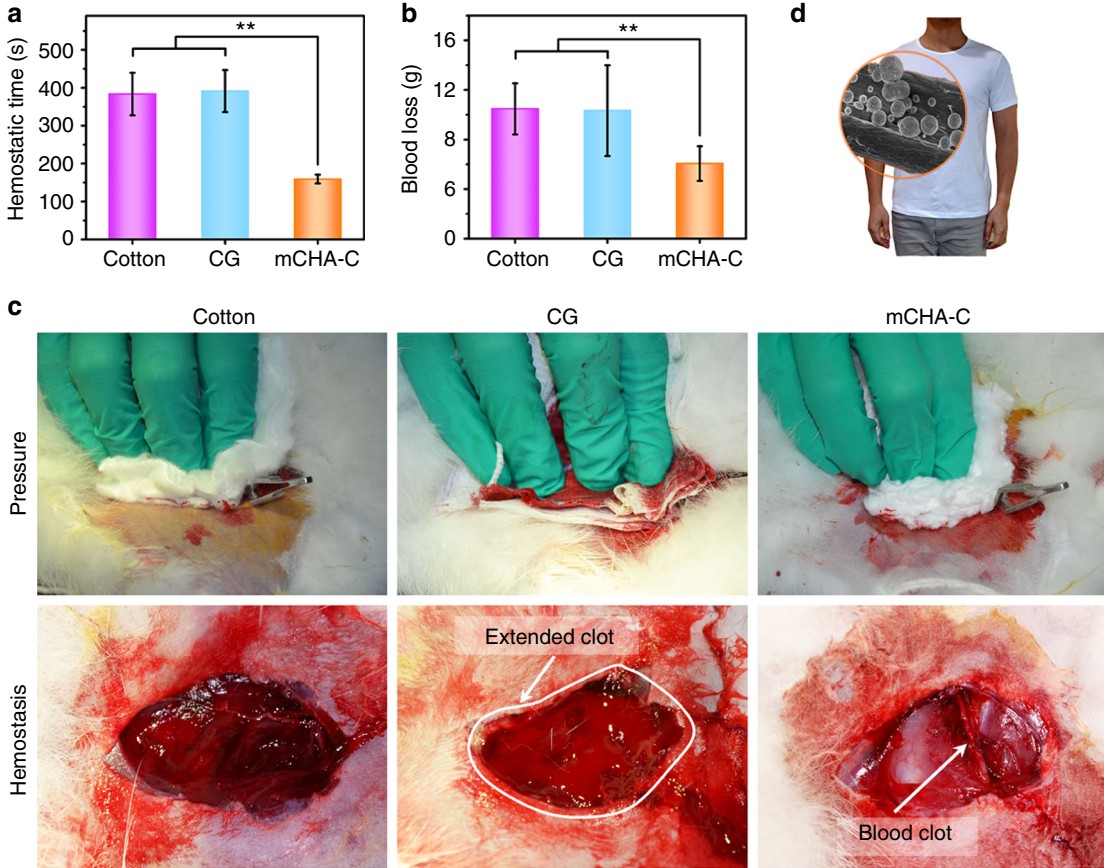

**Fig. 4** In vivo hemostatic capacity evaluation of hemostats. Quantitative analysis of hemostatic time (**a**) and blood loss (**b**) in the rabbit femoral artery injury model. Source data are provided as a Source Data file. Data values corresponded to mean ± standard deviation (SD), $n = 8$. Error bars represent SD. **$P < 0.01$, one-way analysis of variance (ANOVA). **c** Hemostasis was assessed upon manual pressure on the rabbit lethal femoral artery injury with cotton (left), CG (mid), or mCHA-C (right). **d** Image of mCHA/T-shirt

during the early stages of the crystallization process and later grew to spherical aggregates binding on cotton fiber tightly. Due to the perfect combination of CHA and cotton, the as-synthesized mCHA-C was further evaluated in both in vitro and in vivo procoagulant activity test by plasma clotting assay and rabbit femoral artery injury model. It exhibits much better performance of short clotting time and less blood loss than CG. In daily life, this material is geared toward topical hemorrhage control application in compressible injuries. Ideally, zeolite immobilized life vest makes it promising in addressing life-threatening injuries where immediate medical care is not possible in low-income countries and war zones, which is a meaningful step forward in the development of on-demand and targeted hemostat.

## Methods

**Materials**. Ludox HS-30 colloidal silica (30 wt% suspension in $H_2O$) was purchased from Sigma-Aldrich (Shanghai, China). Sodium hydroxide (NaOH), aluminium hydroxide (Al(OH)$_3$), calcium chloride (CaCl$_2$), and ethanol were obtained from Sinopharm Chemical Reagent Co., Ltd (Shanghai, China). The porcine plasma was obtained from Yuhang District Slaughter House (Hangzhou, China). Combat Gauze (Z-Medica, USA) was purchased from commercial source. Degreased cotton (material: 100% cotton; quality: high absorbent and soft; absorbency rate: 3–5 s; making standard: USP international standard; whiteness: 80%), which was purchased from Jiaozuo Union Medical Materials Co., Ltd. It is already known that the natural cotton is soaked in alkaline solution, boiled, bleached, washed, and dried to prepare this material. The degreased cotton has considered OH groups. T-shirts made of cotton were purchased from Xiamen Xinhong Extension Trade Co., Ltd. The T-shirt must be consisted of 100% cotton cellulose and be pretreated with alkaline boiling. For any state of cotton, non-cellulose compounds that deposit on the surface of the cotton fiber, such as lignin, wax, and pectin, should be removed to give access to the interaction between zeolite precursor and cellulose in the cotton fibers.

**Preparation of mCHA-C**. The mCHA-C was prepared from cotton and a gel precursor with molar composition: 9 Na$_2$O:0.7 Al$_2$O$_3$:10 SiO$_2$:331 H$_2$O. In detail, the gel precursor was prepared from two mixtures denoted A and B. Mixture A was prepared by dissolving sodium hydroxide and aluminium hydroxide in deionized water. Mixture B was prepared by mixing of sodium hydroxide, 30 wt% colloidal silica, cotton, and deionized water. Mixture A was added dropwise into Mixture B; during addition, Mixture B was stirred vigorously. The hydrothermal crystallization was performed at 100 °C for 24 h in a Teflon-liner stainless-steel autoclave. Then the cotton composite was immersed in 5 M calcium chloride solution at room temperature for 12 h twice. The sample was washed in deionized water for several times and sonicated for 5 min to remove physisorbed or other loosely bonded zeolite crystals, and finally dried at 65 °C.

**Preparation of Im-zeolite-C**. The in-solution product (zeolite) in the mCHA-C synthesis was collected, and impregnated on cotton as a referential sample. In detail, the zeolite (50 mg) was dispersed in 10 mL of ethanol to prepare the ~`zeolite suspension that was simply dropped onto cotton (150 mg) and then dried. The TGA revealed that the average loading content of zeolite on cotton from three different parts was about 23.5 wt%, which was similar to that of mCHA-C (23.3 wt%).

**Characterizations**. The powder X-ray diffraction (XRD) patterns were recorded on a Rigaku Ultimate IV with Cu Kα radiation (10º·min$^{-1}$). The accelerating voltage and the applied current were 40 kV and 40 mA, respectively. The morphologies of prepared samples were observed via the field-emission scanning electron microscope (FE-SEM, Hiachi SU8010, Japan). The high-resolution transmission electron microscopy (HRTEM) analysis was conducted on FEI Titan G2 electron microscope equipped with an image corrector at 300 kV. Image processing and projected potential simulation were carried out using CRISP and QSTEM codes, respectively[37,38]. To make the mCHA-C hybrid possible to be imaged, mCHA was taken off by Misonix Sonicator S-4000 Ultrasonic Processor (600 W). Elemental analysis of zeolite on cotton was conducted by X-Ray Fluorescence (XRF) spectrometer (ARL ADVANT'X IntelliPower TM 4200, ThermoFisher, USA). Those elements concentration under 0.2 wt% were ignored. The thermogravimetry analyses (METTLER, TGA/DSC 1/1100, Switzerland) were recorded under dynamic oxygen flow by heating the samples to 800 °C at a rate of 10 °C·min$^{-1}$. Nitrogen adsorption isotherms were measured at −196 °C on a Micromeritics ASAP 2020 (V4.01) adsorption analyzer.

**Binding strength between solid content and fabric**. The sonication treatment was used to evaluate the binding strength between solid content and fabric by an ultrasonic cleaner (KS-250D, 250 W, Ningbo Haishu Kesheng ultrasonic equipment Co. Ltd., Ningbo, China). The samples were added into deionized water and treated with sonication at different time intervals. Before the thermogravimetry analysis, all the samples were washed with deionized water for three times and dried in an oven at 65 °C.

**In vitro plasma clotting assay**. An in vitro plasma clotting assay was used to evaluate the procoagulant activity of hemostats[39,40]. The assay measured the coagulant response in terms of clotting time, defined as the time required from activation of the intrinsic pathway of coagulation cascade to the appearance of a firm clot which stuck to the wall of a polystyrene tube. The recalcification of plasma was always applied to ensure a common zero clotting time. In a typical assay, 15 mg of hemostats were presented in a 2 -mL polystyrene tube, and then 1 mL of citrated porcine plasma with 0.2 M calcium chloride solution was injected. The clotting assays were all carried at 37 °C, and the corresponding clotting time was recorded.

Comparison of hemostats before and after washing. The hemostatic materials (15 mg) were added and immersed into 1 mL of deionized water for 1 min, and then taken out from the water. The samples were washed for three times and dried in an oven at 65 °C. The as-prepared samples were then presented in 2- mL polystyrene tube, and the rest steps were completely identical to the measurement of pre-washed hemostats.

Comparison of hemostats before and after sonication. The sonicated hemostatic materials were isolated from detached zeolite or clay before they were measured in in vitro plasma clotting assay. The hemostatic dressings were first added into deionized water, treated with sonication for 10 min, then washed with deionized water for three times and finally dried in an oven at 65 °C. The as-prepared samples were then presented in a 2 -mL polystyrene tube, and the rest steps were completely identical to the measurement of pre-sonicated hemostats.

**In vivo fluorescence imaging**. The hemostats (100 mg) were fixed with 1 mg mL$^{-1}$ Cy5 for 24 h at 4 °C, and were rinsed with water. Due to high loss of kaolin powder, kaolin on the CG was collected in water and labeled with Cy5. Then Cy5-labeled kaolin was simply dropped onto CG gauze to prepare impregnated sample.

ICR female mice with body weight of 20–25 g were obtained from Laboratory Animal Center of Zhejiang University. The study adhered to the Guide for the Care and Use of Laboratory animals[41], and conformed to the animal welfare guidelines of the laboratory animal center of Zhejiang University. All animal operations were in accord with all relevant ethical regulations for animal testing and research. The study received ethical approval from Zhejiang University Experimental Animal Ethics Committee. Mice were fasted for 24 h, but allowed free access to water before the experiments. Following anesthetization, a $1.5 \times 1.5$ cm$^2$ full-thickness wound was punched onto the back of mice. The wound was treated by hemostats and the loss of kaolin or zeolite from hemostats on the wound site was monitored in vivo fluorescence imaging (IVIS Spectrum, excitation: 640 nm, emission: 680 nm). All the mice were finally euthanized with an overdose of sodium pentobarbital.

**Hemostasis in rabbit femoral artery injury**. The hemostatic performance was evaluated in a rabbit femoral artery injury model[42]. The study adhered to the Guide for the Care and Use of Laboratory animals[41], and conformed to the animal welfare guidelines of the laboratory animal center of Zhejiang University. All animal operations were in accord with all relevant ethical regulations for animal testing and research. The study received ethical approval from Zhejiang University Experimental Animal Ethics Committee. Twenty-four female New Zealand White rabbits with body weight of about 2 kg were obtained from Laboratory Animal Center of Zhejiang University. Animals were fasted for 24 h before the experiments, but allowed free access to water. All rabbits were randomly divided into three groups, and each group contained eight rabbits. Before the experiments, the hemostats were dried at 65 °C in an oven for 2 h and sterilized under UV irradiation for 2 h.

Typically, all the rabbits were anesthetized with 40 mg·kg$^{-1}$ sodium pentobarbital through an injector placed in an ear vein. The right femoral artery was exposed from the surrounding issues. A severe arterial hemorrhage was produced by transecting the femoral artery partially (ca. 50% of its circumference), and free bleeding was allowed for 30 s (pretreatment blood loss). Then the blood was wiped by weighed medical gauze from the inguinal cavity before the hemostats (3 g) were used on the wound with manual compression. The compression was interrupted after 2, 5, 7, and 9 min to check for the hemostasis. When the hemostasis was achieved, the time was recorded immediately. Then, these hemostats were collected, and the absorbed blood was recorded as posttreatment blood loss. In addition, an infrared radiation thermometer (FLUKE 59, USA) was applied to record the temperature of wound site. The survival of injured rabbits was observed within 2 h. After being observed for 2 h, the rabbits were finally euthanized with an overdose of sodium pentobarbital.

**Statistical analysis**. The data were presented as mean ± SD. The between-group differences were compared using one-way analysis of variance (ANOVA). $**P < 0.01$ was considered to be greatly significant.

**Reporting summary**. Further information on research design is available in the Nature Research Reporting Summary linked to this article.

## Data availability

All the data supporting the findings of this study are available within the article and its Supplementary Information files or from the corresponding authors upon reasonable request. The Source Data underlying Figs. 3b, d, 4a and b; Supplementary Figs. 1a and b, 2b, 6, 7, 9, 10, 11a, 12, 14, 15, 16, 18, 19, 20 and 21 are provided as a Source Data file.

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

## Acknowledgements

This work was supported by National Natural Science Foundation of China (91545113, 91845203), China Postdoctoral Science Foundation (2017M610363, 2018T110584), Shell Global Solutions International B. V. (PT71423, PT74557), Fok Ying Tong Education Foundation (131015), and the Science & Technology Program of Ningbo (2017C50014). Y. Zhu acknowledges the financial support from the National Natural Science Foundation of China (Grant no. 21771161), the Thousand Talents Program for Distinguished Young Scholars and KAUST Visiting Researcher Scholar. We thank Dr. Zhangxiong Wu for for helpful discussion.

## Author contributions

J.F., Y.Z. and L.Y. conceived the project, designed the experiments, and wrote the paper. J.F. and L.Y. carried out the synthesis, characterizations, and hemostatic evaluation. Y.Z. performed the high-resolution TEM characterizations. X.S., H.C. and L.X. helped with the synthesis and animal experiments. All the authors discussed the results and contributed to the preparation of the paper.

## Additional information

**Competing interests:** J.F., L.Y., L.X., H.C. and X.S. are named on a Chinese patent application (application number: 201810625854.7) and an international patent application (application number: PCT/CN2019/082931) relating to this work. The remaining authors declare no competing interests.

