## [Peer Review File · Nature Communications]

Reviewers' comments:

Reviewer #1 (Remarks to the Author):

The authors present a very interesting research study on 'on site' grown mesoporous chabazite zeolite particles integrated into cotton fibre substrates (termed mCHA-C in the manuscript) for potential 'topical' hemostat application. Inorganic mineral impregnated hemostats are promising technologies for hemorrhage control in topical application on bleeding wounds, and some technologies (e.g. QuikClot, Combat Gauze etc.) have translated into clinical applications in civilian and military scenarios. Considering the fact that traumatic hemorrhage continues to be a life-threatening challenge, technology development to mitigate this risk has significant clinical promise. Thus the scientific premise of the reported research is excellent.

The scientific and technological strength of the reported research is the development of the mCHA-C system and its detailed physico-chemical characterization that establishes its strong integration to cotton fibers, de novo growth of zeolite crystals on the cotton fiber, stable integration of the microscope mesoporous crystalline zeolite aggregates on the cotton that significantly enhances stability, and transformation of this system into hemostats dressings.

While the above aspects signify the strength and promise of this research, several aspects of evaluation are lacking or unclear:

1. Page 3, sentences 68-72 seem to be compound sentences, that need to be revised.
2. A full name of the abbreviation FAU needs to be stated.
3. The authors show that post-sonication the clotting time for mCHA-C remains similar to pre-sonication, whereas for CG and Im-Zeolite-C the clotting time increases in post-sonication compared to pre-sonication indicating dislodgment of hemostatic zeolite from dressing matrix. The experimental description for this as well as the results prods multiple questions:
 - a) From the experimental description it seems that clotting assays were done in a 'closed container' set-up (polystyrene tube) with plasma. In such set up, even if zeolite granules come off the matrix due to sonication, they will still be available in the tube for hemostatic effect. So it is unclear why CG and Im-Z-C re losing their overall hemostatic output post-sonication. Or were all samples sonicated first, then residual hemostatic material isolated (how?) and then used in clotting assay? The experimental descriptions majorly lack clarity in this matter.
 - b) Pre-sonication mCHA-C, CG and Im-Z-C all have comparable clotting time effect. Sonication is not a factor that is realistically relevant in hemostatic mitigation of a hemorrhaging patient. So, if pre-sonication all three groups show comparable clotting time, what is functional benefit of the mCHA-C design over the other two in the context of pro-coagulant activity?
 - c) Cotton itself is also an absorbent (i.e. can concentrate coagulation factors) and can also itself activate coagulation factors. In comparing mCHA-C with CG or Im-Z-C, how is the contribution of the cotton mass (or volume) itself kept constant? That is, do all comparison group have equivalent amount of non-zeolite component?
4. Thermal effect of mCHA-C should be characterized. The authors state that compared to other zeolites a significantly small amount of CHA imparts equivalent hemostatic efficacy and therefore simply based on 'lower amount' it should have 'lower risk of thermal damage'. However, this risk factor needs to be properly characterized.
5. For the femoral artery model, manual pressure was applied. This study should contain a control of manual pressure with just cotton or just gauze (no zeolite), since applied pressure itself is capable of reducing bleeding (stypsis). Also, the 'blood loss' and 'clotting time' parameters would be better to be moved to the main figure and not supplemental figure. Was survival difference analyzed in these studies?

Reviewer #2 (Remarks to the Author):

The topic is really interesting. In my opinion, since this paper is related to zeolite-cotton hybrid, there are some papers that should be considered. It has to be pointed out that some natural zeolites have highly appreciated properties and can be activated and as such implemented in textile for medical purpose. Some references are in supplementary file.

Also, the reference of making composites from zeolite on cellulose are not included (in suppl)

Regarding the paper - it is well presented. I think that textile material should be better described. The state of cellulose material related to its adsorbency and active groups depend on the cotton pretreatment. At least the state should be analyzed and explained in material section or in discussion part.

Please see attached file for more detailed review.

Reviewer 2- Attached file

The topic is really interesting. In my opinion, since this paper is related to zeolite-cotton hybrid, there are some papers that should be considered. It has to be pointed out that some natural zeolites have highly appreciated properties and can be activated and as such implemented in textile for medical purpose, eg.

1. Martin-Kleiner, Irena; Flegar-Meštrić, Zlata; Zadro, Renata; Breljak, Davorka; Stanović-Janda, Silvana; Stojković, Ranko; Marušić, Maruška; Radačić, Marko; Boranić Milivoj. The effect of the zeolite clinoptilolite on serum chemistry and hematopoiesis in mice. // Food and Chemical Toxicology. 39 (2001) , 7; 717-727;
2. Ivković, Slavko ; Deutsch, Ulrich ; Silberbach, Angelika ; Walraph, Erwin; Mannel, Marcus: Dietary Supplementation with the Tribomechanically Activated Zeolite Clinoptilolite in Immunodeficiency, Advances in Therapy, 21 (2004) 2, 135-147
3. Tarbuk, Anita; Grancarić, Anamarija; Šitum, Mirna. Skin Cancer and UV Protection. // AUTEX research journal. 16 (2016) , 1; 19-28
4. Tarbuk, Anita; Grancarić, Anamarija; Magaš, Saša. Modified Cotton Socks – Possibility to Protect from Diabetic Foot Infection. // Collegium antropologicum. 39 (2015) , 1; 177-183
5. Grancarić, Anamarija; Tarbuk, Anita; Kovaček, Ivančica. NANOPARTICLES OF ACTIVATED NATURAL ZEOLITE ON TEXTILES FOR PROTECTION AND THERAPY. // Chemical Industry & Chemical Engineering Quarterly. 15 (2009) , 4; 203-210

Also, the references related to preparing composites from zeolite on cellulose are not included and the paper is related exactly to this problematic:

1. Bischof Vukušić, Sandra; Flinčec Grgac, Sandra; Katović, Drago; Katović, Andrea. SEM Characterisation of the Cellulose Material treated with Polycarboxylic Acid and Zeolite Nanoparticles. // Materials Science Forum. 700 (2012) ; 203-206
2. Flinčec Grgac, Sandra; Katović, Andrea; Katović, Drago. Method of preparing stable composites of a Cu-aluminosilicate microporous compound and cellulose material and their characterisation. // Cellulose. 22 (2015) 3; 1813-1827

Regarding the paper - it is well presented. Only remark I have is related to the textile material used. It should be better described because the state of cellulose material have direct influence to adsorption and bonding. The number of active groups depend on the cotton pretreatment, therefore the cotton in T-shirt, gauze and degreased cotton is quite different... At least the

authors should analyze the state of material in regard of impurities, active groups and explain it in material section or in discussion part.

Reviewer #3 (Remarks to the Author):

The present study reports the synthesis of chabazite (CHA-type) on cotton fibers and their further use for haemostatic material.

The synthesis of zeolites on different type of fibers including cotton and cellulose was reported as well as the factors controlling the growth process. The authors report the synthesis of low-silica CHA-type, a zeolite which does not differ in properties and synthesis conditions from other low silica zeolites as LTA-type and FAU-type synthesized on vegetal fibers. It should be noted that the water capacity, the property important to stop the hemorrhagy, of LTA and FAU zeolites is even higher than that of CHA-type.

The authors use a gel yielding zeolite Y (FAU-type) and indeed they get this zeolite in the bulk. I guess the synthesis of CHA on the fibers was an accident, which is not a problem if this result can be properly explained. However, the particular effect of the fiber surface leading to changing zeolite type is not explained. Instead there is some attempt to explain the crystallization mechanism citing the well known formation of warm-like gel particles (reported by a number of authors: Subotic, Rimer, ..) which does not provide any valuable information.

I should mention that the presence of textural mesopores in agglomerates of tiny crystals is more than expected.

Briefly, I don't see a novelty in the preparation of cotton-zeolite composite that justifies the publication of the paper in a generalist journal.

Additional comments:

- Figure 1: the projections presented in i-p images are useless for this study. It is not clear what the authors want to show them, since the XRD proves the presence of CHA.

- The stability test is not convincing. The authors compare a tectosilicate (zeolite) which is grown on the fibers with layered silicate (clay) that is ex situ deposited, the later is certainly not an appropriate reference.

- P. 3 (bottom) the superior performance of CHA in respect to LTA is based on literature data.

Again, the water capacity of LTA is not lower that the CHA zeolite. What is the explanation of this result?

In conclusion, I recommend rejection.

Referee # 1

Comments:

The authors present a very interesting research study on 'on site' grown mesoporous chabazite zeolite particles integrated into cotton fibre substrates (termed mCHA-C in the manuscript) for potential 'topical' hemostat application. Inorganic mineral impregnated hemostats are promising technologies for hemorrhage control in topical application on bleeding wounds, and some technologies (e.g. QuikClot, Combat Gauze etc.) have translated into clinical applications in civilian and military scenarios. Considering the fact that traumatic hemorrhage continues to be a life-threatening challenge, technology development to mitigate this risk has significant clinical promise. Thus the scientific premise of the reported research is excellent.

The scientific and technological strength of the reported research is the development of the mCHA-c system and its detailed physico-chemical characterization that establishes its strong integration to cotton fibers, de novo growth of zeolite crystals on the cotton fiber, stable integration of the microscope mesoporous crystalline zeolite aggregates on the cotton that significantly enhances stability, and transformation of this system into hemostats dressings.

While the above aspects signify the strength and promise of this research, several aspects of evaluation are lacking or unclear:

1. Page 3, sentences 68-72 seem to be compound sentences, that need to be revised.

Response:

We appreciate very much the reviewer's valuable comments and suggestions. In the revised manuscript, the compound sentences have been corrected to "we propose a successful design of zeolitic hemostat that delivers a fast, long-lasting and safe treatment of hemorrhage. The design requires: i) a proper selection of zeolite with a porous network enabling fast diffusion and adsorption of adsorbate molecules; ii) the zeolite immobilization that allows a long-lasting application and prevents the leaching problem; iii) a non-toxic and low-cost hemostatic system that is easy to operate" (Page 3).

2. A full name of the abbreviation FAU needs to be stated.

Response:

Thanks for the suggestion. Faujasite, the full name of FAU has been added in the revised manuscript (Page 9).

3. The authors show that post-sonication the clotting time for mCHA-C remains similar to pre-sonication, whereas for CG and Im-Zeolite-C the clotting time increases in post-sonication compared to pre-sonication indicating dislodgment of hemostatic zeolite from dressing matrix. The experimental description for this as well as the results prods multiple questions:

- a) From the experimental description it seems that clotting assays were done in a 'closed container' set-up (polystyrene tube) with plasma. In such set up, even if zeolite

granules come off the matrix due to sonication, they will still be available in the tube for hemostatic effect. So it is unclear why CG and Im-Z-C re losing their overall hemostatic output post-sonication. Or were all samples sonicated first, then residual hemostatic material isolated (how?) and then used in clotting assay? The experimental descriptions majorly lack clarity in this matter.

Response:

Thanks for raising this question. The detailed experimental description of post-sonication and isolation has been added into the revised manuscript (see Page 16). All the residual hemostatic materials were isolated from dislodged zeolite or clay before they were measured in the plasma clotting assay. In detail, the hemostats were first added into deionized water, treated with sonication for 10 min, then washed with deionized water for three times and finally dried in an oven at 65 °C. The dislodged zeolite or clay were removed by the deionized water.

b) Pre-sonication mCHA-C, CG and Im-Z-C all have comparable clotting time effect. Sonication is not a factor that is realistically relevant in hemostatic mitigation of a hemorrhaging patient. So, if pre-sonication all three groups show comparable clotting time, what is functional benefit of the mCHA-C design over the other two in the context of pro-coagulant activity?

Response:

Thanks for raising this question. In our study, the plasma clotting assay was done in a closed polystyrene tube. As we mentioned in the manuscript, even if CG or Im-zeolite-C was soaked into water, considerable loss of kaolin or zeolite particles was observed (Fig. 3a). For pre-sonication hemostats, the active component is still available in the tube for hemostatic effect even though they fall off from the matrix in the plasma clotting assay. Thus, CG, Im-zeolite-C and mCHA-C have similar pro-coagulant activity in our *in vitro* plasma clotting assay.

We agree with the referee that sonication itself is not a factor directly relevant in hemostatic mitigation. Nevertheless, it works as an effective tool to evaluate the binding strength between the active hemostatic component and substrate as well as the overall mechanical stability of the hybrid hemostat under harsh conditions. This seemingly overshoot stability test is actually very important for a hemostat to meet practically the diverse complicated hemorrhage conditions, during transportation, storage and application. The sonication treatment has also been widely accepted as a routine hash test to evaluate the mechanical stability of composite materials [Adv. Mater. 13, 1491-1495 (2001)].

Moreover, the stability of hemostat against the water flow is also an important issue, because in practice massive blood will flow out from the vessel in a traumatic injury. Especially for main artery, the massive bleeding will result in considerable loss of loosely-attached kaolin or zeolite from the matrix. To mimic this condition, we washed the hemostats with deionized water for three times to evaluate the functional benefit of mCHA-C over the other two hemostats in the context of pro-coagulant activity. As shown in Fig. 3b and Fig. S12, there is obvious decrease of the pro-coagulant activity of CG and Im-zeolite-C due to the loss of their active

component. In contrast, the mCHA-C design maintains procoagulant activity after washing.

The above discussion has been added into the revised manuscript (Fig. 3b in the main manuscript and Fig. S12 in the supplementary information).

Fig. 3b Plasma clotting time of hemostats before and after washing with deionized water; data values corresponded to mean \pm SD, $n = 3$. ****** $P < 0.01$, one-way analysis of variance (ANOVA).

Figure S12. The relative residual weight of hemostatic component after washing with

deionized water; data values corresponded to mean \pm SD, n = 3. **P < 0.01, one-way analysis of variance (ANOVA).

In addition to the revised Fig. 3b, we have also added the following text in main manuscript (Page 16).

The hemostatic materials were washed with deionized water for three times and dried in an oven at 65 °C. The as prepared samples were then presented in 2 mL polystyrene tube, and the rest steps were completely identical to the measurement of pre-washed hemostats.

c) Cotton itself is also an absorbent (i.e. can concentrate coagulation factors) and can also itself activate coagulation factors. In comparing mCHA-C with CG or Im-Z-C, how is the contribution of the cotton mass (or volume) itself kept constant? That is, do all comparison group have equivalent amount of non-zeolite component?

Response:

Thanks for raising this question. Im-zeolite-C was prepared with an equivalent zeolite loading amount compared to mCHA-C (23.5 wt% vs 23.3 wt%). These two samples have equivalent amount of non-zeolite component (cotton). For the Combat Gauze (CG, Z-Medica), a composite of aluminosilicate (kaolin) and gauze, the thermogravimetry analysis (TGA) revealed that the loading content of kaolin on gauze was 10-25 wt%, indicating a non-uniform distribution. In our study, the total mass of three hemostats were kept constant in plasma clotting assay and rabbit femoral artery injury model.

4. Thermal effect of mCHA-C should be characterized. The authors state that compared to other zeolites a significantly small amount of CHA imparts equivalent hemostatic efficacy and therefore simply based on 'lower amount' it should have 'lower risk of thermal damage'. However, this risk factor needs to be properly characterized.

Response:

Thanks for raising this question. The thermal effect of mCHA-C has been characterized by monitoring the temperature of the wound site in rabbit femoral artery injury model. The temperature at wound site treated with mCHA-C was not statistically different from normal temperature (Fig. S18). Hence the mCHA-C had no exothermic reaction in the rabbit femoral artery injury model. In the revised manuscript, the statement has been corrected to “in addition, mCHA-C overcame the exothermic reaction during the application of zeolite-based granular hemostats (Ref. 16, 36 and 37)”.

Figure S18. Thermal effect of cotton, CG and mCHA-C in the rabbit femoral artery injury model.

5. For the femoral artery model, manual pressure was applied. This study should contain a control of manual pressure with just cotton or just gauze (no zeolite), since applied pressure itself is capable of reducing bleeding (stypsis). Also, the 'blood loss' and 'clotting time' parameters would be better to be moved to the main figure and not supplemental figure. Was survival difference analyzed in these studies?

Response:

Thanks for these suggestions. As suggested by the referee, a control of cotton with manual pressure has been conducted. The clotting time, blood loss and images of cotton in rabbit femoral artery injury have been presented in the revised manuscript (Fig. 4a,b,c). The mCHA-C showed significantly higher blood-clotting capacity and lower blood loss, compared to the cotton group ($P < 0.01$).

Also, the blood loss and clotting time parameters for the femoral artery model have been moved to the main manuscript (Fig. 4a,b). The survival analysis of hemostatic agents have been presented in Fig. S19. The rabbit femoral artery injury model is fatal, which would result in 100% mortality without treatment (excessive bleeding). The application of cotton and CG with manual pressure could mitigate massive hemorrhage, but the survival rate was 75% and 87.5% owing to more blood loss, which is the important lethal factor in injured trauma. Moreover, the mCHA-C offered a statistically significant advantage by increasing the survival rate to 100% with effective hemostasis and less blood loss.

Figure 4. Quantitative analysis of hemostatic time (a) and blood loss (b) in the rabbit femoral artery injury model.

Figure 4c. Hemostasis was assessed upon manual pressure on the lethal femoral artery injury with cotton (left), CG (mid) or mCHA-C (right).

Figure S19. Survival analysis of rabbits treated with each hemostatic dressing in the fatal femoral artery injury model.

Referee # 2

Comments:

1. The topic is really interesting. In my opinion, since this paper is related to zeolite-cotton hybrid, there are some papers that should be considered. It has to be pointed out that some natural zeolites have highly appreciated properties and can be activated and as such implemented in textile for medical purpose. Some references are in supplementary file.

Response:

Thanks for raising this suggestion. As suggested by the reviewer, we have cited the recommended references in the revised manuscript. Please see Ref. 11-15.

2. Also, the reference of making composites from zeolite on cellulose are not included (in suppl)

Response:

Thanks for raising this suggestion. As suggested by the referee, we have cited the recommended references in the revised manuscript. Please see Ref. 21 and 29.

3. Regarding the paper - it is well presented. I think that textile material should be better described. The state of cellulose material related to its adsorbency and active groups depend on the cotton pretreatment. At least the state should be analyzed and explained in material section or in discussion part.

Response:

Thanks for raising this question. The cotton used in our study was degreased cotton (material: 100% cotton; quality: high absorbent and soft; absorbency rate: 3-5 sec; making standard: USP international standard; whiteness: 80%), which was purchased from Jiaozuo Union Medical Materials Co., Ltd. It is already known that the natural cotton is soaked in alkaline solution, boiled, bleached, washed and dried to prepare this material. The degreased cotton has considered OH groups. T-shirts made of cotton were purchased from Xiamen Xinhong Extension Trade Co., Ltd. The T-shirt must be consisted of 100% cotton cellulose and be pretreated with alkaline boiling. For any state of cotton, non-cellulose compounds that deposit on the surface of the cotton fiber, such as lignin, wax and pectin, should be removed to give access to the interaction between zeolite precursor and cellulose in the cotton fibers. In the revised manuscript, the description of textile material has been added to the methods section.

Referee # 3

Comments:

The present study reports the synthesis of chabazite (CHA-type) on cotton fibers and their further use for haemostatic material.

The synthesis of zeolites on different type of fibers including cotton and cellulose was reported as well as the factors controlling the growth process. The authors report the synthesis of low-silica CHA-type, a zeolite which does not differ in properties and synthesis conditions from other low silica zeolites as LTA-type and FAU-type synthesized on vegetal fibers. It should be noted that the water capacity, the property important to stop the hemorrhagy, of LTA and FAU zeolites is even higher than that of CHA-type.

Response:

We thank the insightful comments from the reviewer. Following these comments, we carefully summarized those representative studies on the topic of zeolite-substrate composites and cited the relative references in the manuscript (Ref. 21-29 and 40). In these studies, there are generally three synthetic strategies towards the corresponding zeolite-substrate composites, including *ex situ* deposition, *in situ* growth and covalent linker binding. **i) *Ex situ* deposition.** Those reported zeolitic composites are usually synthesized simply by using impregnation or physical mixing methods (Ref. 26-29), involving little or weak interactions between zeolites and substrates. It severely restricts the effective loading of zeolites as the active components for hemostatic applications and results in significant leaching problems. **ii) *In situ* growth.** Most *in situ* grown zeolite-substrate composites are composed of vegetable fibers coated with the LTA or FAU zeolites, by directly immersing vegetable fibers into the synthesis gel (Ref. 21-25). This allows the conformal growth of a polycrystalline zeolitic shell onto the vegetable fiber (**Fig. Xa**), where the interaction among the zeolitic crystallites within the shell is much stronger than that between the shell and fiber. Accordingly, the mechanical properties of the composite are actually determined by the brittle zeolitic shell rather than flexible vegetable fiber, which would cause remarkable damage of the zeolitic surface-lining of the composite and dislodgment of zeolite films from fibers once bended or twisted [Adv. Mater. 13, 1491-1495 (2001)] and thus restricts its hemostatic applications, especially in treating main artery injuries. In addition, the mechanical properties of the as-grown composite prohibit the scalable and facile fabrications of diverse textile-based wearable system for a wide range of applications associated with the early hemorrhage control. **iii) Covalent linker binding.** There are studies that found zeolite crystals can be readily and strongly attached to the surfaces of vegetable fibers *via* covalent linkages (Ref. 40). The resulting zeolite-tethering vegetable fibers retain flexibility during subsequent manipulations without loss of significant amounts of zeolite crystals even after sonication. However, the incorporation of considerable amount of binders not only largely dilutes the zeolites in the composite as the active component but also significantly reduces their accessible surface area by blocking the micropores [Microporous Mesoporous Mater. 55, 93-101 (2002)]. Moreover, another potential drawback of the covalent linkages or other binders lies in the unavoidable leaching of

the binders themselves.

In this work, we reported a distinct synthetic strategy that allows the direct “on-site” growth of chabazite zeolite particles onto cotton fibers without using any binders. The individual mCHA particles are tightly-bonded with the cotton substrate through chemical bonding, which ensures the high loading and low leaching of mCHA active components in the zeolite-cotton hybrid hemostat. At the same time, unlike the composite hemostats composed of vegetable fibers coated with a continuous conformally grown zeolitic shell, the mechanical properties of the cotton fibers in the as-synthesized mCHA-C hybrid hemostat are well maintained (**Fig. Xb**), which allows the scalable and flexible fabrication of diverse shape-adaptive textile-based wearable devices for early hemorrhage control. More importantly, these CHA particles are single crystalline and hierarchically structured with both mesopores and micropores. Although the mCHA zeolite itself does not surpass the FAU or LTA zeolites in terms of the equilibrium water absorption capacity, the hierarchically porous structure of mCHA zeolites allows a faster water absorption rate that is more important for the early stage hemorrhage control. In this response, we carried out a dynamic water vapour sorption measurement over mCHA and FAU zeolites, which clearly shows that the mCHA zeolites exhibit faster adsorption and diffusion rate of water molecules than FAU zeolites (**Fig. Y**). Especially in the first few minutes, the relative water vapour uptake of mCHA zeolite shows almost 1.26 times as high as that of FAU zeolites. This feature of mCHA manifest itself a superior hemostat for early hemorrhage control. In this revision, we have emphasized these points accordingly.

Figure X. (a) SEM image of a zeolite Y-linen composite fiber [J. Porous Mater. 3, 143-150 (1996)] and (b) mCHA-C composite in our study.

Figure Y. Relative adsorption kinetic curves of water vapor over the mCHA and FAU in 30% RH at 298 K.

The authors use a gel yielding zeolite Y (FAU-type) and indeed they get this zeolite in the bulk. I guess the synthesis of CHA on the fibers was an accident, which is not a problem if this result can be properly explained. However, the particular effect of the fiber surface leading to changing zeolite type is not explained. Instead there is some attempt to explain the crystallization mechanism citing the well known formation of warm-like gel particles (reported by a number of authors: Subotic, Rimer, ..) which does not provide any valuable information.

I should mention that the presence of textural mesopores in agglomerates of tiny crystals is more than expected.

Briefly, I don't see a novelty in the preparation of cotton-zeolite composite that justifies the publication of the paper in a generalist journal.

Response:

Thanks for raising this question. We agree that the phase formation mechanism of the CHA zeolites is very important for this study. As is well known, diverse zeolitic structures can be synthesized *via* organic or inorganic structure-directing agents (SDA) or SDA-free routes such as seeded-growth under hydrothermal conditions [Microporous Mesoporous Mater. 82, 1-78 (2005)]. Actually, different types of zeolites may crystallize from gels with slightly different compositions or under varied reaction conditions, and they could even exhibit interzeolite transformation *via* either SDA or SDA-free routes, as long as their crystal structures can be well registered to each other upon a common composite building unit (CBU). In our experiment, both CHA and FAU zeolites are observed in solution while only CHA particles grow on the cotton fibers. This indicates both CHA and FAU zeolites could nucleate and crystallize from the as-derived gel particles. Because CHA and FAU zeolites have a common *d6r* CBU and their interconversion is widely reported

[Chem. Mater. 27, 2056-2066 (2015)], the growth environment plays a decisive role in directing the formation of different zeolite phases. Therefore, the FAU zeolites with a lower density (13.3 T atoms per nm³) tend to nucleate and crystallize from those low-density amorphous gel particles in solution. In contrast, the CHA zeolites with a higher density (15.1 T atoms per nm³) tend to nucleate and crystallize from those gel particles directly grown on the cotton fibers. These fibers are made of dense-phase silicates where the silicate lattice puts on a strong constraint and direct the nucleation and crystallization of the denser CHA phase. In this revision, we further proved the essential role of those gel particles anchored on cotton fibers in directing the CHA structure by growing zeolites on cotton fibers from pre-formed gel particles in solution (see supplementary information). It is found that only FAU zeolites crystallize from those pre-formed gel particles in solution while no zeolite particles are observed on cotton fibers. These facts unambiguously support our proposed growth mechanism of CHA zeolites on the cotton fibers. The gel particles formed on the cotton surface at the early-stage could further self-assemble and crystalize into mesoporous CHA zeolitic crystals probably via surface migration (Fig. 2c). Additionally, due to the much higher framework density of CHA zeolites compared with the gel particles bonded with cotton fibers, the nucleation and crystallization of CHA particles from these gel particles are accompanied with the concurrent formation of mesopores. Following the reviewer's suggestions, the above discussions on the mechanistic aspects of CHA growth are included in this revision.

Additional comments:

- Figure 1: the projections presented in i-p images are useless for this study. It is not clear what the authors want to show them, since the XRD proves the presence of CHA.

Response:

We thank the reviewer for these comments. We totally agree that XRD alone already allows the unambiguous crystalline phase identification of the mCHA zeolites. XRD only provide long-range ordered structural information averaged over the whole coherence length. While HRTEM and electron diffraction (ED) allow site-specific and orientation-specific structural elucidations. For example, the HRTEM and ED patterns taken along two distinct projections are important to provide local microstructural information that are however invisible for XRD. The corresponding observation directly unravels the single-crystal nature of mCHA zeolite, of which the mesoporosity originates from the oriented-assembly and crystallization of CHA nanocrystals. As widely reported [J. Am. Chem. Soc. 2014, 136, 2503–2510], the creation of mesopores that interrupts the microporous CHA framework without reducing its overall crystallinity accounts for both the high stability of the hierarchical structure and the fast molecular diffusion within it. It is distinct from most mesoporous zeolites synthesized using soft-templating methods and featured for their polycrystalline pore walls and poor stability.

- The stability test is not convincing. The authors compare a tectosilicate (zeolite)

which is grown on the fibers with layered silicate (clay) that is ex situ deposited, the later is certainly not an appropriate reference.

Response:

Thanks for raising this question. In this work, we intend to compare the performance of as-synthesized mCHA-C hybrid with a commercialized hemostat manufactured by the cutting-edge technology. The Combat Gauze (CG), a composite of aluminosilicate (kaolin) and gauze, is the first-line CoTCCC-recommended hemostatic dressing (CoTCCC, the U.S. Department of Defense Committee on Tactical Combat Casualty Care), which is the state-of-the art commercial product to address compressible massive bleeding. The comparison between mCHA-C and CG manifests the former hybrid material one of the most prospective next-generation hemostats towards the high performance, low-cost, scalable and flexible applications for the early hemorrhage control in our daily life.

- P. 3 (bottom) the superior performance of CHA in respect to LTA is based on literature data. Again, the water capacity of LTA is not lower that the CHA zeolite. What is the explanation of this result?

Response:

We agree with the reviewer that the overall water absorption capacity of LTA is not lower that the CHA zeolite. In the revised manuscript, the sentence of “chabazite-type (CHA) zeolites usually exhibit excellent dehydration activity that is an important property for blood concentration and hemostasis, which is comparable with many other zeolites such as zeolite A” has been revised accordingly. For the early hemorrhage control, faster adsorption and diffusion rate of water molecules in the hemostat is another highly demanded property. As is discussed previously, the introduction of mesoporosity into the CHA zeolites even further increases the pore sizes, interrupts the microporous framework of zeolites and results in shorter diffusion length, all of which together contribute the faster diffusion and absorption of water molecules as well as the more effective concentration of blood components at the hemorrhagic site.

Editorial Note: Reviewer #3 was unable to review the revised manuscript; as such, the other reviewers were asked to comment on if the authors had satisfactorily addressed the issues raised by reviewer #3. In comments to the editor it was stated the issues of reviewer #3 had been addressed.

REVIEWERS' COMMENTS:

Reviewer #1 (Remarks to the Author):

The authors have adequately responded to majority of the critique comments, by carrying out additional experiments and associated data analysis. This has significantly improved the depth of information provided in the manuscript.

The authors are commended on incorporating additional details regarding:

1. Stability of the zeolite hemostat when exposed to water
2. Methodology for the polystyrene tube based 'stability and activity' experiments.
3. Control experiments for exothermic effect.
4. Supplementary Information adding details to the materials characterization.

One minor comment only:

In the introduction or end discussion/conclusion, the authors should clearly state that the technology/material is geared towards 'topical hemorrhage control application in compressible injuries', and distinguish the fact that the technology is not geared towards non-compressible hemorrhage and blunt trauma (with internal damage). This does not undermine the hemostatic potential of the technology/material but rather emphasizes that this could be a complimentary or adjunctive system to transfusion-based mitigation of traumatic hemorrhage when both compressible and non-compressible injuries are present.

Reviewer #2 (Remarks to the Author):

The paper has been significantly improved. All the question and remarks were included and corrected.

REVIEWERS' COMMENTS:

Reviewer #1:

The authors have adequately responded to majority of the critique comments, by carrying out additional experiments and associated data analysis. This has significantly improved the depth of information provided in the manuscript.

The authors are commended on incorporating additional details regarding:

1. Stability of the zeolite hemostat when exposed to water
2. Methodology for the polystyrene tube based 'stability and activity' experiments.
3. Control experiments for exothermic effect.
4. Supplementary Information adding details to the materials characterization.

One minor comment only:

In the introduction or end discussion/conclusion, the authors should clearly state that the technology/material is geared towards 'topical hemorrhage control application in compressible injuries', and distinguish the fact that the technology is not geared towards non-compressible hemorrhage and blunt trauma (with internal damage). This does not undermine the hemostatic potential of the technology/material but rather emphasizes that this could be a complimentary or adjunctive system to transfusion-based mitigation of traumatic hemorrhage when both compressible and non-compressible injuries are present.

Response:

We sincerely thank the referee's insightful comments. We have added the following text in the conclusion.

"In daily life, this material is geared towards topical hemorrhage control application in compressible injuries."

Reviewer #2:

The paper has been significantly improved. All the question and remarks were included and corrected.

Response:

We sincerely thank the referee for recommending the publication of our paper. We hope this work could contribute to developing on-demand and targeted hemostat, addressing life-threatening injuries where immediate medical care is not possible.